# Body Fat Is Superior to Body Mass Index in Predicting Cardiometabolic Risk Factors in Adolescents

**DOI:** 10.3390/ijerph20032074

**Published:** 2023-01-23

**Authors:** Núbia de Souza de Morais, Francilene Maria Azevedo, Ariane Ribeiro de Freitas Rocha, Dayane de Castro Morais, Sarah Aparecida Vieira Ribeiro, Vivian Siqueira Santos Gonçalves, Sylvia do Carmo Castro Franceschini, Silvia Eloiza Priore

**Affiliations:** 1Department of Nutrition and Health, Universidade Federal de Viçosa, Viçosa 36570-900, Brazil; 2Graduate Program in Public Health, Universidade de Brasília, Brasilia 70910-900, Brazil

**Keywords:** adiposity, cardiometabolic risk, nutritional status, principal components analysis, X-ray absorptiometry

## Abstract

Background: Excess adiposity is one of the main risk factors for the development of cardiovascular and metabolic diseases. The purpose of this study is to compare cardiometabolic risk factors in eutrophic adolescents with a high body fat percentage (%BF) with eutrophic adolescents with adequate %BF and those with excess weight and %BF. Methods: Cross-sectional study with 1043 adolescents. This study presented power equal to 99.75%. Body fat and anthropometric, clinical and biochemical indicators were evaluated. Participants were grouped according to body composition classified by body mass index (BMI) and body fat percentage. Statistical analyses were performed using R software version 4.0.2, adopting a significance level of 5%. The Mann–Whitney test, principal components analysis and logistic regression were performed. Results: It was observed that the SG was more similar to GC2 than to GC1 for both sexes, demonstrating that there was a greater similarity between these groups in relation to the evaluated factors. Higher values for TC, SBP and TG were associated with the SG when the CG1 was used as reference, controlled for sex and age. Likewise, higher TC values and lower levels of SBP, TG and LDL were related to SG when the CG2 was used as reference. Conclusion: Body fat assessment is more effective in predicting risk factors and cardiometabolic diseases than BMI alone.

## 1. Introduction

The behavioral changes that occur in adolescence can favor the adoption of a sedentary lifestyle and unhealthy diet, which, associated with genetic and physiological factors, can contribute to the development of overweight and obesity [1]. Changes in body adiposity throughout adolescence are physiological and important for the growth and development of these individuals, but, in excess, it can be a risk factor for cardiometabolic diseases [1,2,3]. 

Excess weight among these individuals is considered a global public health problem, and the prevalence of obesity increased tenfold from 1975 to 2016 [4]. The diagnosis of obesity is performed, as recommended by the World Health Organization, by the thresholds of the body mass index (BMI) [5]. However, although less invasive, low cost and widely used to identify excess body weight, BMI is not able to distinguish lean mass from fat mass, which can lead to underestimating or overestimating obesity. Thus, an adolescent classified as eutrophic by BMI may have a high percentage of fat and be at greater risk of developing cardiometabolic complications [6,7].

Adolescents with excess body adiposity have higher chances of developing arterial hypertension, insulin resistance and dyslipidemia since excess adiposity is one of the main factors involved in the development of cardiovascular and metabolic diseases [3]. The obesity developed in adolescence can be maintained or worsened over the years, and an adolescent with obesity can become an adult with obesity with a higher risk of morbidity and mortality. Thus, high adiposity is understood as one of the main health problems in adolescents of both sexes worldwide [8,9].

Understanding how the body composition of adolescents impacts cardiometabolic risk factors is important in the context of public health to enable the construction of viable strategies for the prevention of chronic diseases in these individuals, in adolescence and also in adulthood. Therefore, the objective of this study is to compare cardiometabolic risk factors in eutrophic adolescents with a high percentage of body fat (%BF) with eutrophic adolescents with adequate %BF and those with excess weight and %BF. We hypothesized that, despite having a normal weight, individuals with excess adiposity may be at high risk of developing risk factors and cardiometabolic diseases.

## 2. Methods and Materials

### 2.1. Study Design

The study was carried out with secondary data from two cross-sectional studies carried out in the city of Viçosa-MG from 2010 to 2015.

The study population consisted of adolescents of both sexes, aged 10 to 19 years, who lived in urban or rural areas of the city when the collections were performed.

### 2.2. Sample Selection

The databases were checked separately until they were included in the study, in order to check whether all the necessary variables were included. After this stage, they were merged, and the adolescents were selected. If any adolescent had been assessed in more than one survey, data from the most recent assessment were maintained. Thus, 1043 adolescents were chosen to be part of this study, 735 females and 308 males.

Those who had anthropometric and body composition measurements, blood pressure results, lipid profile, fasting glucose and insulin available in the databases used were included.

The power of the study was calculated using the OpenEpi^®^ program, online (www.OpenEpi.com accessed on 12 November 2021), considering as exposure the excess body fat (greater than or equal to 25% for girls and 20% for boys) and as an outcome high triglyceride levels. The calculation was based on the frequency of changes in triglyceride values in the group of adolescents with excess body fat (25.6%) and with adequate fat percentage (13.7%). A power equal to 99.75% was obtained.

### 2.3. Characterization of the Population

The adolescents were divided into groups. The study group: SG—eutrophy by BMI and high %BF, and the comparison ones: CG1—eutrophy by BMI and adequate %BF and CG2—excess weight by BMI and high %BF.

### 2.4. Evaluated Variables

#### 2.4.1. Anthropometric and Body Composition Assessment

All anthropometric measurements were performed by researchers who were trained during the pilot studies before data collection began. The same protocols, instruments and devices were used for anthropometric and body composition measurements in both studies; therefore, the information collected is homogeneous, which guarantees the relevance of the results found in this research.

For weight measurement, an electronic digital scale (Kratos^®^, São Paulo, Brazil) was used, with a maximum capacity of 200 kg. Weighings were performed according to the proposal of the World Health Organization (WHO) [10]. For greater reliability of the measurements, the scale was calibrated with a standard weight. A portable stadiometer (Alturexata^®^, Belo Horizonte, Brazil) with an extension of up to 2.20 m was used to measure height. The measurements were performed following the techniques recommended by the WHO [10].

Waist (WC) and hip (HC) perimeters were measured using a flexible and inelastic measuring tape (Cardiomed^®^, São Luis, MA, Brazil), with a maximum extension of 2 m, taking care not to compress the tissues. Measurements were taken in duplicate, assuming a maximum variation of 0.5 cm, and in case of a result greater than this value, a third measurement was performed, using the average between the two closest measurements. The WC was measured at the midpoint between the lower margin of the last rib and the iliac crest, in the horizontal plane [11]. The HC was measured in the gluteal region, encircling the largest horizontal portion between the waist and the knees [12].

The body mass index (BMI) was calculated (WHO, 1995) and classified according to age and sex (BMI/A) using the software Who AnthroPlus, in z-score values, with ≥1 being considered overweight [13].

Waist–hip (WHR) and waist-to-height (WHtR) ratios were calculated, and for the WHR classification, values ≥0.50 were considered as the cutoff point for the presence of abdominal obesity, regardless of age and sex [14].

Dual energy X-ray absorptiometry (DXA) equipment (Lunar Prodigy Advance DXA System version: 13.31, GE Healthcare, Madison, WI, USA) was used to estimate the percentage of total body fat (%BF), android (%) and gynoid (%), as well as the fat of the trunk (%), arms (%) and legs (%), separately. This is considered the gold standard for assessing body composition. The examination was performed in the morning, with the adolescents fasting for 12 h and following the evaluation protocol [15]. The %BF was classified as high when greater than or equal to 25% for girls and 20% for boys [16].

#### 2.4.2. Clinical and Biochemical Parameters

The cardiometabolic risk factors considered in this study were blood pressure, total cholesterol (TC), high-density lipoprotein (HDL), low-density lipoprotein (LDL), triglycerides (TG), fasting glucose and insulin resistance (IR).

Blood pressure was measured and classified according to the recommendations of the Brazilian Society of Cardiology [17], considering age, sex and height percentile [18].

Biochemical analyses were performed in an accredited laboratory. The individuals were instructed to fast for 12 h before collection. Samples were collected by venipuncture in the morning.

The classification of serum lipid values was performed according to the VII Brazilian guideline on dyslipidemia and prevention of atherosclerosis [18].

Fasting glucose was classified according to the Brazilian Society of Diabetes [19]. The IR was evaluated by the mathematical model HOMA–IR (homeostasis model assessment—insulin resistance), based on insulin and fasting glucose levels. The cutoff points considered indicative of the presence of IR followed the proposal contained in the guidelines of the Brazilian Society of Diabetes [19].

#### 2.4.3. Cardiometabolic Risk Factors

The following cardiometabolic risk factors were considered: excess adiposity in the different locations evaluated (android, gynoid, legs, arms and trunk), altered levels of systolic and/or diastolic blood pressure, total cholesterol, LDL, HDL, TG, glucose, presence of insulin resistance, high WHR and WHtR.

### 2.5. Statistical Analysis

The databases were created using Excel software; these data were double entered independently by two researchers and were validated in the Epi Info software. Statistical analyses were performed with the aid of software R version 4.0.2, adopting a significance level of 5%.

The Kolmogorov–Smirnov [20] test was performed, by which it was found that the variables did not present a normal distribution.

The Mann–Whitney test was performed to compare the numerical variables between the study group (SG) and each of the comparison groups (CG1 and CG2) separately, according to sex.

Principal components analysis (PCA) was performed to identify the factors associated with each group (SG, CG1 and CG2) [21]. For this analysis, the adolescents were divided according to nutritional status groups, and all cardiometabolic risk factors were considered as variables.

To determine the odds ratio (OR), a bivariate exploratory analysis was carried out in order to identify the variables of interest. Therefore, the existence of an association between the body composition groups with each variable considered a cardiometabolic risk factor was determined by bivariate logistic regression. Variables that presented *p* < 0.20 [22] in the bivariate analysis were included in the multiple model. For the analyses, age and sex were used as adjustment variables.

Then, multiple logistic regression was performed, in which, at each step, the variables that did not significantly change the OR and the confidence intervals (*p* < 0.05) were discarded until a final model was obtained, using the backward method. The quality of the model was evaluated by the information criterion of Akaike and the adjustment by the test of Hosmer and Lemeshow, from the function hoslem.test<t7/>, which indicates good fit when *p* > 0.05 [23].

### 2.6. Ethical Aspects

All studies whose databases were included in this study were submitted and approved by the Ethics Committee in Research with Human Beings of the Federal University of Viçosa (CEP/UFV), when they were carried out and at the time the databases were merged in 2018 (opinion No. 2.879.661).

The participants of all the surveys and their guardians were informed about the objectives of the same and provided informed consent (IC), in the case of adolescents aged 18 or 19, or assent (TA) for those under 18 years of age when the data were collected. Only adolescents who delivered the duly signed terms were included in the surveys.

## 3. Results

In the study group (SG) (eutrophy due to BMI and high %BF) 90.8% (*n* = 373) were female, a similar situation was observed in the CG2 group (high BMI and %BF), with 72.9% (*n* = 153) girls. In the eutrophic group (CG1), 49.5% (*n* = 213) were female.

Regarding girls, it was observed that almost all variables differed between the SG and each of the comparison groups, with the exception of HDL and HOMA-IR, which did not differ between SG and CG1, and WHR, TC between SG and CG2 (Table 1).

Considering male adolescents, age, TG, TC and LDL were different between SG and CG1. WHR, TC, LDL, HDL, TG and glucose measurements did not differ between SG and CG2 (Table 1).

Girls and boys in the SG had, on average, a BMI of 2.6 kg/m^2^ (*p* < 0.0001) and 1.5 kg/m^2^ (*p* = 0.003) higher, respectively, as well as 6.5 cm (*p* <0.0001) and 5.03 cm (*p* = 0.002) more waist circumference, compared to normal-weight individuals with adequate BF% (results not shown in the table).

The PCA showed the distribution of cardiometabolic risk factors among the nutritional status groups, according to the similarity between the groups. Dimensions 1 and 2 together explained 46.6% of the distribution of these variables for females and 49.1% for males (Figure 1).

Analyzing the numbers, it was observed that the SG was more similar to GC2 than to GC1, for both sexes, demonstrating that there was a greater similarity between these groups in relation to the evaluated factors. The size of the arrows for each risk factor indicates its influence on the disposition of individuals in the group to which it points (Figure 1).

When we observe the cluster graph, for both sexes, we can say that the variables that most influenced the separation of CG1 were leg fat and HDL; the other factors point to the direction in which SG and CG2 are disposed, demonstrating similarity between these groups regarding these factors (Figure 1).

Logistic regression showed that overweight adolescents classified by BMI were more likely to have elevated SBP and HOMA-IR, as well as lower HDL. Adolescents with obesity according to the %BF were more likely to have HOMA-IR and DBP levels (Appendix A).

In relation to body composition groups, the final regression model showed that higher values of TC, SBP and TG were associated with the SG when the CG1 was used as reference, controlled for sex and age. Likewise, higher TC values and lower levels of SBP, TG and LDL were related to SG when the CG2 was used as reference. The other risk factors evaluated were not included in the multivariate model, as there was no significant difference between the SG and the other groups (Table 2).

The analysis of adjustment adequacy of the models showed a value greater than 0.05, according to the test of Hosmer and Lemeshow.

## 4. Discussion

This study showed that most adolescents classified as having excess BF, regardless of BMI, were female. In addition, the results showed that female adolescents with excess weight and body fat had higher medians for most of the biochemical factors evaluated and lower for HDL, demonstrating that body composition is related to changes in blood pressure, lipid profile, fasting glucose and insulin resistance. Nevertheless, a similar result was observed for boys, with the exception of WHR and TC, which showed higher medians in the SG.

Principal component analysis (PCA) was performed to identify factors associated with each group. This analysis showed that, for both sexes, there was a greater similarity between the SG and CG2 in relation to the evaluated risk factors.

The final regression model showed that higher values of TC, SBP and TG were associated with the SG when the CG1 was used as reference. Likewise, higher TC values and lower levels of SBP, TG and LDL were related to SG when the CG2 was used as reference.

In this context, it is known that during adolescence there is a physiological increase in body fat and that it occurs more intensely in females. This increase in BF is essential for the growth and maintenance of menstrual cycles [24]; however, in excess, it is a risk factor for the development of obesity. In boys, the superior increase in muscle mass predominates [25].

In this phase, growth and development occur quickly and intensely [26]. However, high consumption of high calorie foods, rich in fats and sugars, and low consumption of those considered healthy, in addition to low levels of physical activity are still common [27]. The physiological changes associated with unhealthy eating habits and a sedentary lifestyle make this population vulnerable to the development of obesity and other risk factors for cardiometabolic diseases [28].

Obesity is a serious public health problem worldwide [4], as individuals are developing this disease at younger ages. Worldwide, the prevalence of obesity in children and adolescents aged 5 to 19 years increased from 0.7% and 0.9% in 1975 to 5.6% and 7.8% in 2016, in girls and boys, respectively, according to BMI [29,30].

Overweight adolescents classified by BMI were more likely to have elevated SBP and HOMA-IR, as well as lower HDL. Adolescents with obesity according to the %BF were more likely to have HOMA-IR and DBP levels. Thus, the increase in the prevalence of obesity in this age group is worrying, as this condition can lead to changes in the lipid profile, with an increase in the levels of total cholesterol, triglycerides, LDL and a reduction in HDL levels, in addition to arterial hypertension, resistance to insulin and diabetes, changes that have already been observed in adolescents [31,32,33,34,35,36]. In addition, excess body fat in adolescence increases the risk of developing cardiometabolic diseases and other morbidities in adulthood, besides being a predictor of increased mortality, mainly due to cardiovascular diseases [37,38,39,40].

Excess body fat is an independent risk factor for cardiometabolic diseases, including in adolescents; therefore, individuals with normal weight and high BF% are at increased risk of developing insulin resistance, dyslipidemia, metabolic syndrome and cardiovascular disease [41], which increases the risk of early mortality [42].

Individuals with excess adiposity have increased cytokine secretion, which can lead to the blocking of insulin receptor signals; this makes the body less sensitive to the insulin produced and causes the condition called insulin resistance [43], in addition to leading to changes in the lipid profile, which results in the development of cardiometabolic diseases [39,40]. Furthermore, these individuals have a greater accumulation of fat in the arteries, which makes the heart have to work harder to pump blood throughout the body, and this causes increased pressure on the inner walls of the arteries. This condition is a risk factor for the development of arterial hypertension [37].

Studies have shown that, despite having adequate weight, an individual may have excess adiposity, and this condition has also been observed in adolescents. Morais et al. [44] observed that, among 274 Brazilian adolescents aged 14 to 19 years, 15% were overweight by BMI; however, 53.9% had excess body fat. Furthermore, among 631 adolescents of both sexes, aged 11 to 18 years, Ripka et al. [45] found that BMI overestimated the %BF in 68.4% of boys (5.0 ± 4.0%) and underestimated it in 67.5% of girls (−3.9 ± 2.6%).

Prado Júnior et al. [46] found that adolescents with excess body fat, although eutrophic by BMI, had a higher prevalence of elevated TC and TG, altered insulin and HOMA, in addition to low HDL, compared to the group of eutrophic adolescents with adequate BF%. Furthermore, the group of adolescents with normal BMI and high %BF showed similar behavior to the group with excess weight and body fat, in relation to fasting glucose and TC. This confirms that body adiposity is more associated with cardiometabolic risk factors, demonstrating the need to assess body composition using specific measures and indexes to estimate body fat.

Olafsdottir, Torfadottir and Arngrimsson [47] evaluated 182 18-year-old adolescents and found that those with normal weight with a high %BF had a higher IR compared to normal-weight adolescents with adequate BF% (*p* = 0.003). Moreover, of five adolescents with metabolic syndrome (MS), four were eutrophic with a high %BF. Furthermore, the study showed that eutrophic adolescents with high %BF were 2.2 times more likely to have one or more risk factors for MS (OR = 2.2; 95% CI: 1.2–3.9) in relation to adolescents who were eutrophic with adequate %BF.

In a longitudinal study carried out for about seven and a half years, Wiklund et al. [48] evaluated 396 girls with a mean age of 11 years at baseline and found that the cardiometabolic risk score was higher in the group of adolescents with normal weight and high %BF, when compared to the eutrophic group with adequate %BF and with low weight.

In a recent study carried out with 1919 Colombian adolescents, the authors found that those with normal weight according to BMI, but with excess body fat, had higher mean levels of waist circumference, triglycerides and cardiometabolic risk score, in addition to a lower mean HDL value. In addition, girls in this nutritional situation had a higher mean systolic blood pressure [7].

It is noteworthy that BMI is the most used indicator to diagnose nutritional dystrophies; however, this index is not able to accurately estimate body adiposity, since it does not allow distinction between muscle mass and fat mass [5,6,7], and especially during adolescence, BMI does not reflect the corporate changes that occur. In this sense, the isolated use of BMI becomes inappropriate [5,6,7].

The principal component analysis performed in this study showed that the EG presented greater similarity with the CG2 in relation to the factors evaluated. A study with 1421 adolescents of both sexes, aged 10 to 13 years, also performed this analysis, and, although it did not specifically target the three nutritional status groups, showed that percent body fat was associated with all cardiometabolic risk factors and carried the strongest loading coefficient [49]. These findings reinforce the notion that excess body adiposity, despite adequate weight, is directly related to the development of cardiometabolic risk factors during adolescence, and the assessment of body composition in these individuals is essential.

It is important to consider that the effect of obesity in adolescents on the risk of cardiovascular diseases in adults may be irreversible even if in adulthood the individual becomes eutrophic [39]. This demonstrates the need for specific health promotion and disease prevention actions during adolescence, such as food and nutrition education actions in schools, since at this age individuals spend most of their day in the school environment. Since it is known that a majority of life habits are defined at this stage of life, it is a favorable period to encourage the adoption of a healthy lifestyle [50,51].

It is noteworthy that this work is relevant, as there are few studies in the literature that assess the presence of cardiometabolic risk factors in eutrophic adolescents with excess body fat, especially comparing those with normal weight by BMI and those with excess weight and adiposity.

Moreover, this research has an expressive sample number and comprises adolescents aged 10 to 19 years and of both sexes. In addition, a method considered gold standard in the assessment of body composition (DEXA) was used.

## 5. Conclusions

The findings of this study show that, despite having a normal weight, individuals with excess adiposity are at greater risk of developing risk factors, as well as cardiometabolic diseases, than those with normal weight and adequate body fat. This demonstrates the importance of examining adiposity in the monitoring of adolescents, in order to obtain a more complete assessment of nutritional status.

Given this context, the assessment and monitoring of nutritional status from an early age is essential to enable the early diagnosis and treatment of obesity and to enable interventions to be more effective in order to prevent diseases in the future.

## Figures and Tables

**Figure 1 ijerph-20-02074-f001:**
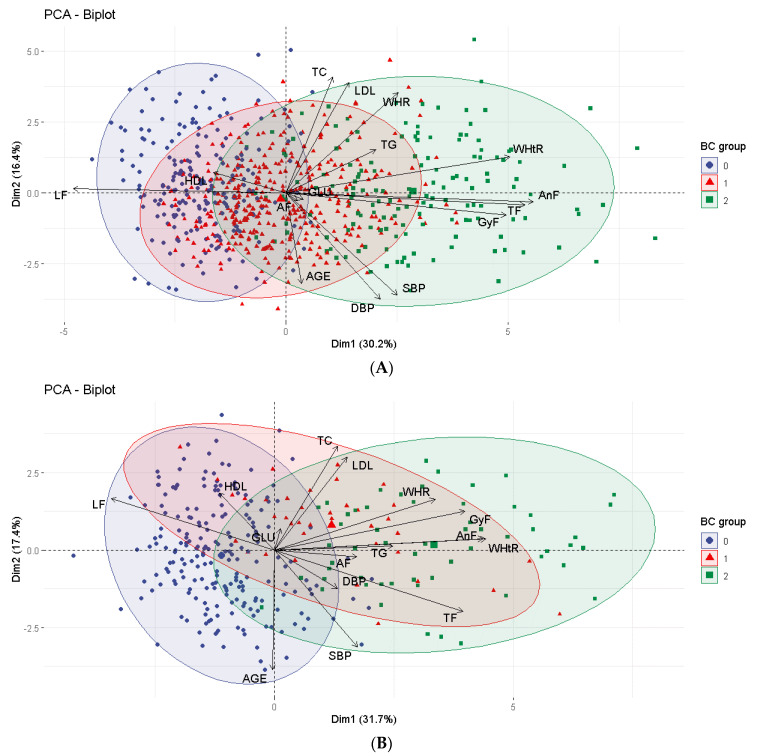
Cardiometabolic risk factors associated with each body composition group, for females (**A**) and males (**B**). Legend: 0—eutrophy by BMI and adequate %BF (CG1); 1—eutrophy by BMI and high %BF (SG); 2—excess weight by BMI and high %BF (CG2). AF: arm fat; AnF: android fat; BC: body composition; BMI: body mass index; DBP: diastolic blood pressure; GyF: gynoid fat; GLU: glucose; HDL: high-density lipoprotein; LF: leg fat; LDL: low-density lipoprotein; SBP: systolic blood pressure; TC: total cholesterol; TF: trunk fat; TG: triglycerides; WHtR: waist–height ratio; WHR: waist–hip ratio.

**Table 1 ijerph-20-02074-t001:** Comparison of the study group with groups 1 and 2, according to sex. Viçosa-MG, Brazil, 2010 to 2015 (*n* = 1043).

Variables	Females (*n* = 735)			Males (*n* = 308)		
	SG (*n* = 373) Median (Min–Max)	CG1 (*n* = 209) Median (Min–Max)	CG2 (*n* = 153) Median (Min–Max)	SG (*n* = 38) Median (Min–Max)	CG1 (*n* = 213) Median (Min–Max)	CG2 (*n* = 57) Median (Min–Max)
Age (years)	16 (10–19) ^† ‡^	15 (10–19)	15 (10–19)	12 (10–19) ^†^	14 (10–19)	13 (10–19)
BMI (kg/m^2^)	20.2 (15.0–24.0) ^† ‡^	17.5 (12.0–23.0)	25.5 (20.0–47.0)	17 .7 (16.0–30.0) ^† ‡^	17.6 (13.0–24.0)	24.2 (19.0–34.0)
BF (%)	29.6 (25.0–42.0 ^† ‡^	20.9 (9.0–25.0)	39.9 (27.0–57.0)	23.0 (20.0–35.0) ^† ‡^	11.3 (5.0–20.0)	30.0 (20.0–49.0)
WHtR	0.4 (0.3–0.5) ^† ‡^	0.4 (0.3–0.5)	0.5 (0.4–0.8)	0.5 (0.4–0.6) ^† ‡^	0.4 (0.3–0.5)	0.5 (0.4–0.6)
WHR	0.8 (0.66–1.0)	0.8 (0.66–1.1)	0.8 (0.7–1.0)	0.8 (0.5–1.0) ^†^	0.7 (0.68–0.99)	0.8 (0.5–1.0)
SBP (mmHg)	101.5 (75.0–134.0) ^† ‡^	97.5 (73.0–150.0)	107.0 (85.0–165.0)	96.7 (80.0–127.0) ^‡^	98.5 (74.0–143.0)	106.0 (85.0–136.0)
DBP (mmHg)	66.0 (47.0–91.0) ^† ‡^	61.0 (44.0–110.0)	68.5 (51.0–100.0)	59.0 (47.0–68.0)	58.0 (40.0–97.0)	61.5 (47.0–74.0)
Total cholesterol (mg/dL)	154.0 (84.0–283.0)	151.0 (46.0–241.0)	155.0 (91.0–239.0)	168.0 (113.0–217.0) ^†^	150.0 (870.0–234.0)	164.0 (90.0–217.0)
LDL (mg/dL)	87.0 (29.0–201.0) ^† ‡^	84.0 (23.0–165.0)	90.2 (40.0–167.0)	96.8 (63.0–136.0) ^†^	87.2 (39.0–156.0)	99.8 (28.0–148.0)
HDL (mg/dL)	52.0 (28.0–161.0) ^‡^	52.0 (26.0–97.0)	45.0 (23.0–100.0)	47.5 (30.0–117.0)	49.0 (29.0–106.0)	45.0 (30.0–71.0)
Triglycerides (mg/dL)	64.0 (28.0–212.0) ^† ‡^	60.0 (26.0–97.0)	76.0 (26.0–272.0)	67.0 (25.0–210.0) ^†^	61.0 (14.0–130.0)	73.0 (55.0–248.0)
Glucose (mg/dL)	85.0 (64.0–408.0)	84.0 (3.0–105.0)	86.0 (65.0–114.0)	85.5 (75.0–101.0)	86.0 (81.00–91.00)	85.0 (70.0–111.0)
HOMA-IR	1.5 (0.0–6.0) ^‡^	1.4 (0.0–4.0)	2.1 (1.0–11.0)	1.4 (0.0–4.0) ^‡^	1.2 (0.0–3.0)	2.4 (1.0–11.0)
Android Fat (%)	18.2 (6.0–49.0) ^† ‡^	8.9 (4.0–21.0)	35.1 (11.0–59.0)	14.7 (7.0–72.0) ^† ‡^	5.5 (4.0–18.0)	26.2 (11.0–50.0)
Gynoid Fat (%)	37.8 (19.5–51.0) ^† ‡^	28.3 (9.0–37.0)	48.0 (33.0–63.0)	31.1 (19.0–55.0) ^† ‡^	16.9 (4.0–33.0)	39.2 (29.0–58.0)
Trunk fat (%)	30.3 (15.3–54.0)	29.7 (17.1–49.4)	29.6 (18.0–51.5)	27.7 (15.2–46.5)	30.3 (16.8–45.2)	31.0 (18.6–42.0)
Arm fat (%)	8.3 (4.0–19.7)	8.2 (4.2–56.6)	8.1 (4.6–11.3)	8.1 (5.7–12.5)	8.3 (4.6–12.4)	8.1 (4.6–11.3)
Leg fat (%)	56.3 (3.3–77.0)	57.5 (22.4–67.9)	56.4 (5.6–69.5)	57.8 (42.1–69.1) ^‡^	56.4 (42.0–72.0)	54.8 (44.4–66.1)

Legend: BMI: body mass index; SG: eutrophy by BMI and high %BF; CG1: eutrophy by BMI and adequate %BF; CG2: excess weight by BMI and high %BF; HDL: high-density lipoprotein; HOMA-IR: homeostasis model assessment—insulin resistance; LDL: low-density lipoprotein; SBP: systolic blood pressure; DBP: Diastolic blood pressure; WHtR: waist–height ratio; WHR: waist–hip Ratio. Mann–Whitney. ^†^ SG × CG1: *p* < 0.05; ^‡^ SG × CG2: *p* < 0.05.

**Table 2 ijerph-20-02074-t002:** Association of cardiometabolic risk factors with the study group (SG). Viçosa-MG, Brazil, 2010 to 2015 (*n* = 1043).

	Model 1 *		Model 2 **	
Risk Factors	Odds Ratio (IC95%)	Value of *p*	Odds Ratio (IC95%)	Value of *p*
TC	1.01 (1.01–1.02)	0.02	1.02 (1.01–1.04)	0.01
DBP	1.05 (1.02–1.07)	0.001	-	-
SBP	-	-	0.93 (0.91–0.94)	<0.001
TG	1.01 (1.01–1.02)	<0.001	0.99 (0.98–0.99)	<0.001
LDL	-	-	0.96 (0.94–0.98)	<0.001

* Model 1: CG1 (eutrophy by BMI and adequate %BF) considered as a reference; ** Model 2: CG2 (overweight by BMI and high %BF) considered as a reference. Legend: TC: total cholesterol; LDL: low-density lipoprotein; DBP: diastolic blood pressure; SBP: systolic blood pressure; TG: triglycerides.

## Data Availability

Not applicable.

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
