# Peer review of "Body Fat Is Superior to Body Mass Index in Predicting Cardiometabolic Risk Factors in Adolescents"

_ijerph, 2023, doi:10.3390/ijerph20032074_

Round 1

Reviewer 1 Report

The objective of this study is to compare cardiometabolic risk factors in eutrophic adolescents with a high percentage of body fat (%BF), with eutrophic adolescents with adequate %BF and those with excess weight and %BF.

It is well-documented that abdominal adiposity increases the risk for cardiovascular disease. What is the novelty of the current study? What additional contributions can this study bring to the literature? Authors need to provide justifications for the current study. 

What does adequate BF mean? 

What are the clinical implications for the current study? 

Author Response

We appreciate the constructive comments that helped us improve the quality of our manuscript. Attached is a file with the responses for each suggested change.

Reviewer 2 Report

Comments were integrated into the manuscript (pdf).

Author Response

(The authors gave the same response as above.)

Round 2

Reviewer 2 Report

1) We used the TG to calculate the power of the study because it was the risk factor with the highest frequency of change in the sample": what do the authors consider a (meaningful) change and what do you mean by "frequency of change"? Power calculation still not clear.

2) About the 0,20 (This cutoff point is used to ensure that any variable that may have an effect on the multiple model is included. The 0.05 point is too low and with it, we could exclude important variables, which can have an effect together with the others): Couldn't understand the explanation.

4) The title does not reflect the study findings.

5) Abstract lacks important methodological and results info. Make sure it included as much relevant info as possible. The abstract should reflect (very briefly) everything that has been done in the study.

6) English writing needs improvement.

Author Response

Thanks again for the constructive comments that helped us to improve the quality of our manuscript. Attached is a file with the responses for each suggested change.
